# Improving Innovation and Access to Combination Vaccines for Childhood Immunization in China

**DOI:** 10.3390/ijerph192315557

**Published:** 2022-11-23

**Authors:** Jiyan Ma, Zhuo Li, Yinuo Sun, Zuokun Liu, Yuan Dang, Yangmu Huang

**Affiliations:** Department of Global Health, School of Public Health, Peking University, Beijing 100191, China

**Keywords:** innovation, access, health disparity, combination vaccine, childhood immunization

## Abstract

Background: combination vaccines can improve timely vaccination coverage and mitigate the social and economic burdens of both caregivers and health systems. Compared to other countries with high immunization performance, China remains behind the curve in promoting the inclusion of new combination vaccines into national vaccination schedules. The domestic research and development pipeline faces many technical obstacles, regulatory pressures, and competitive opposition. In addition to this, health disparities regarding combination vaccines exist in each dimension of access and their determinants, including availability, accessibility, acceptability, and quality. Our study aims to provide a cross-disciplinary analysis of China’s combination vaccines (from innovation to access) and identify the main factors that affect the attitudes and behavior choices for combination vaccines. Method: systematic reviews and secondary data analysis will be conducted to map the landscape of combination vaccines in China and the determinants influencing their innovation and access. A cross-sectional survey will be performed in seven provinces of China based on geo-economic representativeness among caregivers with children that are between 2 and 24 months old and are registered in the national immunization system. Questionnaires will be used to examine the relationship between each dimension of access and their determinants. These questionnaires will cover the caregivers’ knowledge, attitude, and willingness to pay for combination vaccines, as well as their perceptions about vaccination services. Semi-structured interviews with the suppliers (public and private) and healthcare providers will help identify research gaps and the key challenges they face when developing and introducing combination vaccines in China. Discussion: using a combined approach, with cross-country and multi-disciplinary support from experts, our research is designed to fill the information gaps in China’s combination vaccine industry across the innovation-access spectrum. It will lead to evidence-based recommendations which will foster greater access to innovation-enhancing combination vaccines for childhood immunization in China. Moreover, the multi-dimensional approach could also be adapted beyond combination vaccines to assess innovation and other public goods for health among disadvantaged groups in the future.

## 1. Introduction

Immunization is one of the most cost-effective strategies to promote child health across socioeconomic groups [1]. A core goal of every country’s national immunization plan is to increase vaccination coverage rates for children. Children under 24 months often receive several vaccines at the same time, which increases the risk of fever and local tenderness. Utilizing a combination vaccine is an ideal approach that allows the simultaneous administration of multiple antigens, resulting in fewer injections, fewer health facility visits, and less chance for a child to develop a fear of needles and pain [2]. This approach can improve the number of parents and children who accept vaccinations and provide opportunities to introduce new vaccinations. It also can reduce management, storage, and transportation costs for immunization systems [3]. Now, the DTP-based combination vaccines, which contain diphtheria toxoid (D), tetanus toxoid (T), and pertussis toxoid (P), have been in common use since the middle of the 20th century to include additional antigens, such as the inactivated polio vaccine (IPV), the haemophilus influenzae type B (Hib) vaccine, and the hepatitis B (HepB) vaccine [4]. The successful development and implementation of the 5-in-1 Pentavalent vaccines (DTP-HepB-Hib) is a great example of this, which has led to more effective vaccinations among 517 million children in the world’s 73 poorest countries [5]. As one of the most far-reaching health interventions, combination vaccines can help increase the immunization coverage rate on a global scale and play a crucial role in achieving sustainable development goals (SDGs) by leaving no one behind [6]. Many international organizations, such as the World Health Organization (WHO), the Global Alliance for Vaccines and Immunization (GAVI), and the United Nations International Children’s Emergency Fund (UNICEF), recommend that countries develop and introduce combination vaccines [7,8,9].

Under the combined implementation of the National Immunization Program (NIP, publicly funded) and non-NIP (privately funded), China has become one of the countries with the highest routine vaccination coverage rates [10]. Over four million deaths have been prevented since the launch of NIP and non-NIP [11]. A 99% decrease in the incidence of 12 vaccine-preventable diseases (VPDs) has been achieved, with more than 300 million cases of polio, pertussis, diphtheria, tetanus, hepatitis B, and measles averted [11]. However, compared to other high-performance countries, which have replaced standalone vaccines with combined alternatives, China remains behind the curve in promoting the inclusion of new combination vaccines into national vaccination schedules [12]. Since the current combination vaccines are mainly imported from overseas, and the price is relatively high, only 3 of the 11 NIP vaccines are combination vaccines [13]. Constrained innovation and unequal access to combination vaccines have led to significant social and economic impacts on the households and immunization systems in China [12].

### 1.1. Constrained Domestic Innovation for Combination Vaccines in China

In recent years, the evolving economy and consumption patterns have contributed to the unprecedented growth of privately funded combination vaccines, both in terms of volume and value [14]. The national batch number has increased from 2.09 million in 2013 to 11.86 million in 2020 [15]. However, China’s ability to develop and produce combination vaccines is still in the early development stages, and their supply of vaccines mainly depends on imports. The imported and domestic combination vaccines accounted for 7.1% and 1.2% of the non-NIP market share, respectively [16].

The licensing and registration of new and innovative combination vaccines are urgently needed to provide sufficient, effective, affordable, and high-quality vaccinations for Chinese children. In 2019, China’s Third Vaccine Administration Law presented suggestions for development plans and arranged the necessary funds to improve research, development (R&D), and access to combination vaccines [17]. However, due to regulatory bottlenecks and a lack of strategic planning, China’s R&D of combination vaccines still faces substantial risks, technical obstacles, regulatory pressure, and competitive opposition [8]. Until recently, most of China’s vaccine candidates were at low levels of combination, such as two-combination or three-combination vaccines.

### 1.2. Unequal Access to Combination Vaccines in China

Constrained upstream innovation has impeded the entire vaccine development process from R&D to manufacturing, distribution, delivery, and use. Disparities regarding combination vaccines in China can be summed up in each dimension of access and their determinants. These dimensions include availability, accessibility, acceptability, and quality.

For availability, the vaccination coverage rate of combination vaccines varies greatly among different regions and by different socioeconomic conditions [18]. Adequate and efficient access is constrained by supply chain vulnerabilities and demand uncertainties. Since China’s combination vaccine market is precariously dependent on an extremely small number of pharmaceutical companies, problems can arise from shortages or delays in vaccine manufacturing, distribution, or delivery. For example, the withdrawal of Sanofi’s 5-in-1 vaccine due to the failure in quality assurance resulted in a long-term vaccine shortage in 2017 [19].

For accessibility, the lack of financial resources and information are the most common challenges faced by caregivers (a family member or paid helper who regularly looks after the child, e.g., parents or grandparents). Limited public control over the pricing of China’s combination vaccines allows pharmaceutical companies to set higher prices, which directly affects the ability of low- and middle-income families to buy combination vaccines [20]. Moreover, the caregivers’ rights to request, receive and impart information from healthcare providers on combination vaccines have been neglected. Information provided by a healthcare professional during vaccination empowers caregivers to be involved in assessing their child’s vaccination options. However, when it comes to combination vaccine introduction and application, there is no regulation on the standard provision of health information.

For acceptability, caregivers have different expectations for NIP and non-NIP vaccines. The combination vaccines are outside the national schedule and may lead many parents to think of them as less recommended or important. A better understanding of caregivers’ main expectations and concerns regarding combination vaccination would help identify valuable strategies to increase vaccination coverage among different populations.

For quality, poor-quality vaccines, counterfeit combination vaccines, inappropriate storage conditions, and control during manufacturing are the main concerns [21]. A high prevalence of low-quality combination vaccines was reported in 2017. Eight out of 36 batches of the 5-in-1 combination vaccines for about 715,000 children did not meet the quality requirements [22]. This eroded public trust in combination vaccines.

### 1.3. Research Gaps on Innovation and Access to Combination Vaccines

Innovation and access to combination vaccines are two continuous phases that closely interact. Both these phases require multi-discipline joint efforts to study them with a holistic and global perspective. The studies are supposed to result in more feasible and effective suggestions for improvements that will help China’s immunization systems meet global standards. However, previous studies have mainly focused on only one aspect of combination vaccines, whether that was safety, effectiveness, a health economic evaluation, or factors influencing vaccination domestically [23,24,25,26]. No existing research has conducted a multi-dimensional analysis of China’s combination vaccines using the innovation-access spectrum and experts from diverse backgrounds. No study has covered the availability, accessibility, acceptability, and quality dimensions while looking at both the supply and demand side of the system. Meanwhile, due to the short history of the development of combination vaccines in China, there is a lack of research on macro-environmental factors, R&D strategies, and public-private partnerships that impact vaccine innovation and access. This is not conducive to a long-run adjustment of the supply of combination vaccines, which is exactly what is needed to meet the demand.

### 1.4. Significance

The increasing use of combination vaccines would allow for the simultaneous administration of multiple antigens, resulting in fewer injections and health facility visits. It would also mitigate the economic burden of caregivers and lower the chance of a child developing a fear of needles and pain. At the same time, increasing the use of combination vaccines would improve society’s timely vaccination coverage and reduce the cost that health systems pay for management, storage, and transportation. Our study is designed to fill the gaps in China’s combination vaccine industry across the innovation-access spectrum using both a multi-discipline and a global perspective. The study will investigate the dimensions and determinants of combination vaccine accessibility. It will also assess the feasibility of new vaccine innovation. This analysis will lead to evidence-based recommendations which will foster greater access to innovation-enhancing combination vaccines that meet the need of China’s children. Moreover, the development of China’s combination vaccines would also contribute to the implementation of SDGs and be beneficial to vulnerable populations in other developing and less developing countries.

## 2. Design

### 2.1. Aim and Objectives

Our study aims to conduct a multi-dimensional and cross-disciplinary assessment of the combination vaccine across the innovation-access spectrum. It aims to identify the main challenges and influencing factors of both suppliers and demanders. Finally, it aims to inform the strategies and policies for improving innovation and access to combination vaccines in China.

The objectives of the project are:To study the R&D landscape of combination vaccine candidates in China and identify the key challenges that public and private suppliers face when developing and introducing combination vaccines.To examine the access to combination vaccines for childhood immunization in each dimension of access and their determinants (i.e., availability, accessibility, acceptability, and quality) and to explore the main factors that affect attitude and behavior choices on the demand side of the product (caregivers).To develop strategies and policy options for fostering innovation and access to combination vaccines that address children’s health needs.

### 2.2. Research Framework

The study is based on the WHO’s AAAQ framework to evaluate four essential standards of healthcare access: availability, accessibility, acceptability, and quality. Compared with other frameworks, this model is widely acknowledged for providing a complex network of multi-stakeholders involved in access to healthcare. It includes suppliers (i.e., government agencies, pharmaceutical companies, research institutes, and healthcare providers) and demanders (i.e., caregivers). Meeting the availability standard means having a sufficient number and the right type of combination vaccines for the children who need them. The accessibility standard measures how affordable the combination vaccines are for people at every socioeconomic level. In this standard, financial accessibility and information accessibility are the most relevant determinants of whether caregivers purchase combination vaccines or not. Financial accessibility can be defined as the relationship between the cost of combination vaccine services and the caregiver’s willingness to pay. The accessibility of information refers to the caregiver’s rights to request, receive, and impart information from healthcare providers. Acceptability is the measure of how responsible healthcare providers are to the social and cultural expectations of the caregivers [27]. Meeting the quality standard requires a scientifically sound combination of products that do not jeopardize children’s health, causing vaccine hesitancy.

A mixed-method approach will be conducted to assess China’s combination vaccines from innovation to access (Figure 1). The approach is as follows: (1) Systematic reviews and secondary data analysis will be conducted to map the landscape of combination vaccines in China and the determinants influencing the innovation and access to combination vaccines, (2) A cross-sectional survey will be performed to examine access to combination vaccines in China’s representative settings. This survey will examine the relationship between each dimension of access and their determinants in terms of their availability, financial accessibility, information accessibility, acceptability, and quality, (3) Semi-structured interviews with suppliers (public and private) and healthcare providers will be conducted to identify research gaps and the key challenges faced when developing and introducing combination vaccines in China, (4) Focus group discussions will be convened for feedback and the exchange of ideas among all stakeholders, decision-makers, and executive entities.

## 3. Methods

### 3.1. Literature Review and Secondary Research

A systematic review will be conducted to provide an overview of the macro-environmental factors which can impact the development and introduction of combination vaccines. The review will also examine the opportunities and challenges caused by political, economic, social, and technological factors. The outputs from the review will provide a fact base for semi-structured interviews with public and private providers. It will also inform the decision-making for the ongoing and future R&D plans.

Search terms will focus on the following: (1) political perspective (government policy, law, and regulation, registration pathway, and funding), (2) economic perspective (economic trends, industry growth, and international trade), (3) social perspective (customer preference, population growth, family structure, and lifestyle trend), and (4) technological perspective (the product development process, R&D capacity, emerging technologies, intellectual property rights, and technology incentives). The scope of the review will include up-to-date peer review journals, reports, and internet articles from PubMed, Medline, Scopus, EMBASE, Web of Science, CNKI, and other databases. Central and local policy documents, reports, and technical guidelines will be retrieved from the official websites of the National Health Commission (NHC), National Medical Products Administration (NMPA), China’s Center for Disease Prevention and Control (China CDC), and other ministries and commissions. This study is limited to combination or multi-pathogen vaccines that incorporate antigens against multiple diseases in one injection. We only focus on the combination vaccine candidates in the R&D pipeline and products that are currently in the non-NIP market, such as the DTaP-IPV-Hib vaccine from Sanofi Pasteur (Lyon, France) and the DTaP-Hib vaccine that was developed by Beijing Minhai Biotechnology (Beijing, China).

The secondary research will be carried out to identify the following: (1) map the R&D landscape of combination vaccine candidates from pre-clinical to clinical development phases, and (2) analyze supply and demand, assess vaccine quality, and compare the price of different combination vaccines. The analysis will help identify the research gaps in the current pipeline and the factors associated with unequal access to combination vaccines in China (in terms of availability, quality, and financial accessibility). We will extract administrative data and cluster survey results from the NMPA databases, the Chinese Central Government Procurement (CCGP), the National Bureau of Statistics, the DHS surveys, the WHO/UNICEF immunization coverage estimates, and the documents from the surveyed percentage of the points of vaccines (POVs).

For availability, supply indicators include the number and percentage of combination vaccines in the following: manufacturing, distribution, and delivery by area, population, and point POVs that have at least one combination vaccine in stock, number, and percentage of vaccine shortage per point, and stock-out duration per point. Demand indicators include the coverage rate of combination vaccines per served population/point and the timeliness of vaccination. The following ranges for supply and demand indicators will be used in the assessment: very low (<30%), low (30–49%), fairly high (50–80%), and high (>80%) [28]. For financial accessibility, the median unit price of the combination vaccines will be compared to the average procurement prices published in the vaccine pricing data of UNICEF. This will result in median price ratios (MPRs). According to the WHO/HAI methodology, an MPR of 1.0 means that the local price is equivalent to the international reference price. The following rages will be used: ideal (<1), rational (≤1.5), and irrational (>1.5). For acceptability, each satisfaction factor will be scaled by the caregivers from 0 to 5. For quality, the study relied on the Medicine Quality Assessment Reporting Guidelines. These state that the quality of the combination vaccines will be assessed based on the number of quality control samples taken for testing annually, the number of annually tested samples that failed to meet quality standards, the qualified rate of samples, and the type of issues observed (e.g., active ingredient assay, impurities screening, pharmaco-technical features).

### 3.2. Questionnaire Survey

The survey is designed to evaluate the caregivers’ knowledge and attitude towards combination vaccines and their willingness to pay for them. Caregivers’ perceptions about the information and vaccination services provided for vaccine uptake will also be evaluated. The survey will help examine the relationship between access (based on finances, the information provided, and acceptability) and the determinants of issues in access. The survey will also explore the main factors that affect the demanders’ attitudes and behaviors.

#### 3.2.1. Survey Design

The survey will include questions related to the following: (1) demographic and socio-economic information (age, education, place of residence, occupation, family income, relationship between children and caregivers, number of children, parenting style, type of health insurance, children’s age, gender, ethnicity, allergy history, general reaction after vaccination, and the history and adverse events of injecting combination vaccines), (2) financial accessibility (average time and transportation cost to POVs, accepted price range for combination vaccines and factors of preferences), (3) information accessibility (knowledge on the functions, the adverse events, the price range of combination vaccines, the information channels and the types of combination vaccines. This also covers satisfaction with the consultation and explanation by providers, and the frequency and method of vaccination appointment), (4) acceptability (perceptions of safety, efficacy, reaction, benefits and side effects of combination vaccines, worth relative to cost, satisfaction with waiting time, service time, vaccination technique, attitude of providers, and cleanliness of facility).

#### 3.2.2. Setting and Participants

The study will be conducted in seven provinces based on geoeconomic representativeness: Beijing (North), Shanghai (East), Hubei (Central), Hainan (South), Jilin (Northeast), Sichuan (Southwest), and Gansu (Northwest). Caregivers are eligible to participate if their children are between 2 and 24 months old and registered in the NIP Information Management System (NIPIMS).

#### 3.2.3. Sampling

The districts (counties) of each city will be classified with one of three levels (high, medium, and low) based on the gross domestic product (GDP) per capita. A stratified random cluster sampling will be adopted to select one district (county) from each level. This will be conducted based on the sampling equation: n = (z^2^ ∗ pq)/d^2^. Z-value is 1.96; p is the proportion of children who have been vaccinated by combination vaccines (we use the average coverage rate of 55% [17]); q is the proportion of unvaccinated persons; d is the difference between the estimated point value and the population mean value and is set as 0.1. As such, in the minimum sampling unit, at least 105 caregivers will be selected from the township. Approximately 3–5 community healthcare centers or POVs will be randomly selected from each township to recruit caregivers who bring their children in for vaccination. It is expected that at least 3600 caregivers will participate in the survey.

#### 3.2.4. Statistical Analysis

Descriptive statistics will be used to summarize the demographic and socio-economic characteristics of both the caregivers and children and their access to combination vaccines through Stata 17 (StataCorp, College Station, TX, USA). Proportion, means, standard deviations, and 95% confidence intervals (CIs) will be used for descriptive analysis. In addition, multivariate logistic regression will be performed to investigate the relationship between the dimension of access and its main factors. The independent variables are the characteristics of health facilities and the selected demographic and socioeconomic variables of the caregivers. The dependent variables are the selected indicators of availability, accessibility, and acceptability. Only variables with *p*-values less than 5% after univariate analysis were entered into the multivariate logistic regression model using the backward stepwise exclusion method.

### 3.3. Qualitative Study

#### 3.3.1. Semi-Structured Interview

We will explore the main facilitators and barriers to innovation and access to combination vaccines from the perspective of the suppliers (public sector, private sector, and healthcare providers). The interview guideline will be developed based on the initial outputs of the literature review and secondary research. The format will be an outline script with a list of open-ended questions. Example questions include the following: (1) what impedes and facilitates the R&D of combination vaccines in China? (2) to what extent does the government incentivize the innovation of combination vaccines? (3) are there any plans to introduce combination vaccines into the NIP? (4) what are the key challenges that impact combination vaccine services and benefits?

The targeted interviewees are public-sector suppliers (government agencies, universities, and public research institutes), private-sector suppliers (pharmaceutical companies) in the current pipeline and market, and healthcare providers (CDCs, community healthcare centers, etc.) from surveyed POVs. Approximately 25–30 public and private suppliers will be contacted via initial contacts and snowballing. At least two healthcare providers will be interviewed in each POV. To ensure accuracy, interviewers will take detailed notes and/or audio recordings during the interview and will check for errors as soon as it is over. The thematic analysis method will be used to review, code, and analyze the content of the semi-structured interviews through NVivo 12. The findings will contribute to the development of evidence-based recommendations for promoting the innovation of and access to combination vaccines in China.

#### 3.3.2. Focus Group Discussion

Stakeholders, decision-makers, and executive entities will be invited to participate in the focus group discussion, which will allow feedback and comments about the final results to be heard. This will help identify R&D priorities and barriers to unequal access. This group will also exchange ideas on the strategies and policy options that can be developed to promote the innovation of and access to combination vaccines in China. The study results will be summarized and distributed to participants before the discussion sessions. Targeted parties include pharmaceutical companies, research institutes, government agencies (e.g., NHC and NMPA), delivery systems for vaccines (e.g., CDC and community healthcare center), professors and/or experts in public health, childhood immunization, pharmaceutical policies, and health services. Approximately 15–20 participants will be purposefully chosen to attend face-to-face meetings. The study results and recommendations will be evaluated by experts in the relevant fields and then developed into policy briefs and reports.

## 4. Discussion

Previous combination vaccine studies have concentrated on discussing the technical challenges and future perspectives on product innovation or have focused on identifying the association between access to vaccines and its influencing factors from the angle of a single perspective, such as caregivers or healthcare providers [23,24,25,26]. No existing research has conducted a multi-dimensional analysis of China’s combination vaccine infrastructure across the innovation-access spectrum, and no research has studied the dimensions of availability, accessibility, acceptability, and quality from both the suppliers’ and the demanders’ perspectives. Using a combined approach, with cross-country and multi-disciplinary support from experts, our research could fill the information gaps in China’s combination vaccine industry across the innovation-access spectrum. Our research could inform the ongoing and future development of combination vaccines, which would help meet children’s needs in China. Moreover, the development of China’s combination vaccines would also be beneficial for vulnerable populations in other developing and less developed countries.

However, our study has several limitations. First, the mixed-method approach may require more resources for data collection, management, and analysis. This would extend the length of the study beyond what was originally planned. It may be difficult and time-consuming to resolve discrepancies that arise in the interpretation of the qualitative and quantitative findings. Second, due to the limitation of data and funding, we can only survey seven provinces of China based on geo-economic representativeness. Our questionnaire needs to be further tested in different settings and populations. In addition, other limiting factors that are commonly associated with self-reporting studies, such as the accuracy of recall and personal bias, could inhibit the survey by affecting the measurements of the caregivers’ knowledge and attitudes. Finally, even though our study aims to promote the innovation of and access to combination vaccines, the combination vaccine itself has several drawbacks. For example, the chemical incompatibility and immunologic interference of combination vaccines could be difficult to overcome. It may not be clear which component of a combination vaccine is responsible for specific adverse events. It can also be difficult to ensure that combing antigens will keep the protection provided by each antigen [2].

There might be some external hurdles that emerge in the execution of this study, such as the changes in related policy, regulations, and standards, personnel adjustment, etc. These changes may exert an unexpected influence on the project. The cooperation and support from both the public and private sectors are crucial to the success of the study’s assessment, such as the accessibility to caregiver participants, participation, and the time input of key personnel (interviewees). The continuing coronavirus disease 2019 (COVID-19) pandemic may cause delays in the surveys, the interviews, and the focus group discussions. To minimize the risk outside our institution, a steering committee will be established to enhance the monitoring of the study’s progress and proactively identify problems. Meanwhile, the project will take a flexible strategy of implementation to be able to adjust the project quickly. We will use modern technology to help streamline communications, surveys, interviews, and group discussions and keep clear, open, and frequent communication with our stakeholders and communities. Our team will hold regular online meetings to follow up on the project and promote trust in our multi-stakeholder partnerships.

## 5. Conclusions

Our study will provide a detailed and cross-disciplinary analysis of China’s combination vaccine infrastructure from innovation to access and identify the main factors that affect attitudes and behavior choices regarding these vaccines. The evidence-based recommendations will foster greater access to innovation-enhancing combination vaccines that will meet Chinese children’s needs. These recommendations will also lead to feasible regulatory approaches to improve the innovation of new products and promote the further inclusion of combination vaccines into the national immunization schedule. In the future, the study’s multi-dimensional approach and methodological techniques could be adapted beyond combination vaccines. This could help assess the innovation of and access to other public goods for health among disadvantaged groups in the future.

Our results will be disseminated to policymakers and other relevant stakeholders through scientific articles, policy briefs, and workshops. At the policy level, roundtable discussions and workshops will raise the focus on the importance of improving innovation and access to combination vaccines in China and the challenges that this process brings. Policy briefs will be submitted through research teams in the hope of including evidence-based suggestions in both national and provincial policy dialogues. At the academic level, three papers will be submitted to both domestic and international peer-reviewed journals. These papers will cover the following topics: “R&D landscape of combination vaccines in China”, “Health access to combination vaccines for childhood immunization in China: a cross-sectional study”, and “Knowledge, attitude and willingness of children’s caregivers for combination vaccination in China”. Meanwhile, our research results will be presented at all relevant conferences at the national, regional, and international levels. This study will not provide direct health and financial benefits to the participants. However, both suppliers and caregivers will participate in a self-evaluation and will be able to reflect on the knowledge, awareness, and practice of combination vaccinations. This will help guide future work and play an active role in promoting the innovation of and access to combination vaccines in China.

## Figures and Tables

**Figure 1 ijerph-19-15557-f001:**
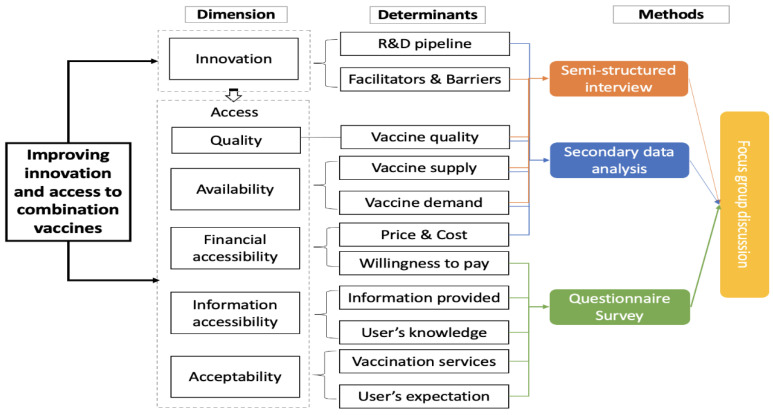
Research framework of innovation and access to combination vaccines.

## Data Availability

No datasets are currently available for this current study.

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
