# Peer review of "Improving Innovation and Access to Combination Vaccines for Childhood Immunization in China"

_ijerph, 2022, doi:10.3390/ijerph192315557_

Round 1
Reviewer 1 Report
Report:
Int. J. Environ. Res. Public Health 2022
Improving Innovation and Access to Combination Vaccines for Children in China.
In this study, Ma et al. intended to provide an overview of a prospective cross-disciplinary analysis of China’s combination vaccines and identify the factors that affect attitudes and behavior choices for combination vaccines. This is an interesting futuristic study, however, the manuscript would benefit from the following corrections/amendments:
1. The authors may like to give reasons why only 3 of 11 NIP vaccines are combination vaccines in China.
2. The researchers may like to give the questionnaires as Supp. data.
3. The authors may like to mention potential hurdles which are likely to emerge in the execution of this prospective study.
4. Limitation of combination vaccines should be discussed.
5. The limitations of the questionnaire Survey should be discussed.
6. The authors claim, "At the academic level, three 369 papers will be submitted to domestic and international peer-reviewed journals...". Please mention the main themes of these prospective papers.
7. The financial and personnel impact of this study is not mentioned.
Reviewer 2 Report
This study is trying to help with vaccine combination in China, especially for children’s needs. The significance of the study is high, since the topic is close related to public health, and many people will be benefited if there’s innovation and improvements. On the other hand, the paper is not perfectly written. Please check the following comments and suggestions.
Major points:
1. For introduction, explain more on why vaccine combination is more important. The current manuscript only has brief description. Moreover, please talk about different kinds of vaccines. For instance, give some examples on what kind of vaccines have the combining potential, and explain why some vaccines can be combined but some cannot. These are all important things you should put in introduction. Generally speaking, the information provided in the introduction part is too superficial. Please add more details.
2. Your article is titled with “children”. However, the whole introduction part didn’t really point out anything closely associated with children. Actually, it is not only your introduction part. Overall, children is seldomly mentioned. If this study is designed based on children, then please add more information and discuss more on vaccination and children. For instance, you mentioned many times “to meet children’s need”. What do children need healthcare-wise?
3. The overall flow of the design and methods sections should be re-organized. Currently, there are a lot of information going back and forth, which is not very clear and logical. If needed, figures and tables may be used to map the workflow.
Minor points:
1. Line 47. How can vaccine combination achieve 14 out of 17 sustainable development goals? This line here needs to have more details to be useful.
2. Line 48-50. Other agencies of the United Nations? For example: … ?
3. Line 53-55. “A 99% decrease in incidence for vaccine-preventable diseases”, like what? Same idea applies to the following line, “a similar decline in mortality and disease burden”, please explain with more details or examples.
4. Also, please check your references. Usually, by the end of each statement, we add references. For the introduction part, line 26-42, line 55-60 and 76-77 needs more references. There are so much more after these. Seems like you either missed some references, or simply wrote a large chunk of texts and then add references. Please go through your entire manuscript and add references accordingly.
Round 2
Reviewer 2 Report
The authors have edited the manuscript based on reviewer comments. These editions have improved the manuscript and brought improvements.
Author Response
Reviewer's comment: The authors have edited the manuscript based on reviewer comments. These editions have improved the manuscript and brought improvements.
- Thanks very much for your comments. Your valued suggestions are very important to this manuscript.